# Impact of the Introduction of a Two-Step Laboratory Diagnostic Algorithm in the Incidence and Earlier Diagnosis of *Clostridioides difficile* Infection

**DOI:** 10.3390/microorganisms10051075

**Published:** 2022-05-23

**Authors:** Nieves Sopena, Jun Hao Wang-Wang, Irma Casas, Lourdes Mateu, Laia Castellà, María José García-Quesada, Sara Gutierrez, Josep M. Llibre, M. Luisa Pedro-Botet, Gema Fernandez-Rivas

**Affiliations:** 1Infectious Diseases Department, Germans Trias i Pujol University Hospital, Carretera de Canyet s/n, 08916 Badalona, Spain; lmateu.germanstrias@gencat.cat (L.M.); saragserra@gmail.com (S.G.); jmllibre.fls.germanstrias@gencat.cat (J.M.L.); mlpbotet.germanstrias@gencat.cat (M.L.P.-B.); 2Faculty of Medicine, Universitat Autònoma de Barcelona, Cerdanyola del Vallès, 08193 Barcelona, Spain; icasas.germanstrias@gencat.cat (I.C.); gfernandezr.germanstrias@gencat.cat (G.F.-R.); 3IGTP, Health Sciences Research Institute Germans Trias i Pujol, 08916 Badalona, Spain; 4Microbiology Department, Clinical Laboratory North Metropolitan Area, Germans Trias i Pujol University Hospital, 08916 Badalona, Spain; jhwang.germanstrias@gencat.cat; 5Preventive Medicine Department, Germans Trias i Pujol University Hospital, 08916 Badalona, Spain; 6Infection Control Nurse, Germans Trias i Pujol University Hospital, 08916 Badalona, Spain; lcastella.germanstrias@gencat.cat (L.C.); mjgarciaq.germanstrias@gencat.cat (M.J.G.-Q.)

**Keywords:** *Clostridioides difficile*, *Clostridioides difficile* toxin, infection control and prevention, healthcare-associated infection, immunosuppressed patients

## Abstract

Our aim was to determine changes in the incidence of CD infection (CDI) following the introduction of a two-step diagnostic algorithm and to analyze CDI cases diagnosed in the study period. We retrospectively studied CDI (January 2009 to July 2018) in adults diagnosed by toxin enzyme immunoassay (EIA) (2009–2012) or toxin-EIA + polymerase chain reaction (PCR) algorithm (2013 onwards). A total of 443 patients with a first episode of CDI were included, 297 (67.1%) toxin-EIA-positive and 146 (32.9%) toxin-EIA-negative/PCR-positive were only identified through the two-step algorithm including the PCR test. The incidence of CDI increased from 0.9 to 4.7/10,000 patient-days (*p* < 0.01) and 146 (32.9%) toxin-negative CDI were diagnosed. Testing rate increased from 24.4 to 59.5/10,000 patient-days (*p* < 0.01) and the percentage of positive stools rose from 3.9% to 12.5% (*p* < 0.01). CD toxin-positive patients had a higher frequency of severe presentation and a lower rate of immunosuppressive drugs and inflammatory bowel disease. Mortality (16.3%) was significantly higher in patients with hematological neoplasm, intensive care unit admission and complicated disease. Recurrences (14.9%) were significantly higher with proton pump inhibitor exposure. The two-step diagnostic algorithm facilitates earlier diagnosis, potentially impacting patient outcomes and nosocomial spread. CD-toxin-positive patients had a more severe clinical presentation, probably due to increased CD bacterial load with higher toxin concentration. This early and easy marker should alert clinicians of potentially more severe outcomes.

## 1. Introduction

*Clostridioides difficile* (CD) is a common cause of healthcare-associated diarrhea, with increasing morbidity and mortality rates and concomitant hospitalization costs [1]. However, community-acquired *Clostridioides difficile* infection (CDI) is rising and currently accounts for 25% to 40% of cases [1]. The incidence of CDI has increased in the last decade, probably due to the growing numbers of patients at risk and improvements in diagnostic methods [2,3]. The incidence of CDI varies across countries, ranging from 2.8 to 15.8 cases per 10,000 patient-days [2,3,4,5]. These differences may be due in part to low suspicion awareness among clinicians and suboptimal diagnostic methods based only on toxin assessment [6].

Reported risk factors for CDI include older age, length of hospitalization, comorbidities and previous exposure to antibiotics and proton pump inhibitors (PPI) [7]. The clinical presentation of CDI varies from mild diarrhea to severe colitis [8]. CDI treatment depends on the severity of disease and the potential risk of recurrence [8]. The cornerstone of the diagnosis has been based on the detection of a toxigenic strain of CD in the stool. However, optimal diagnostic algorithms are still the subject of studies and not all centers offer the same panel of diagnostic tests with nucleic acid amplification test (NAAT) alone or a 2- or 3-step testing with CD glutamate dehydrogenase (GDH) + toxin and optional subsequent NAAT [9]. The existence of asymptomatic carriers with positive NAAT stool tests further complicates the diagnosis of CDI. A stepwise diagnosis protocol including toxins and NAAT could capture CDI cases not identified by CD toxin–enzyme immunoassay (EIA) and shed additional light on the severity of CDI outcomes through a surrogate estimation of CD bacterial load [10].

A surveillance study in Catalonia showed that the application of a multimodal training strategy that included an online course on CDI, two face-to-face workshops, and dissemination of recommendations on its prevention and the use of an optimal diagnostic algorithm significantly increased the incidence of CDI from 2.20 cases per 10,000 patient-days in 2011 to 3.41 in 2016 [11]. An optimal diagnostic algorithm comprises a test with high sensitivity as a screening method, such as GDH, followed by a more specific test to detect toxins either by EIA in a first step or by molecular NAAT methods in a second step.

The aim of the present study was to analyze the changes in the incidence of CDI in a university hospital over a decade with the introduction of a two-step laboratory diagnosis algorithm. Moreover, we analyzed epidemiological characteristics, risk factors, clinical presentation, diagnostic test, treatment and outcome of CDI patients diagnosed in this period.

## 2. Methods

### 2.1. Study Design and Patients

This observational retrospective study was conducted at a 550-bed university-affiliated public hospital located in Badalona, Spain. All adult patients (≥18 years old) with a first episode of diarrhea diagnosed as CDI during the period 1 January 2009–31 July 2018 were included. Diarrhea was defined as ≥3 unformed stools in a 24 h period and symptoms lasting ≥ 24 h. A CDI case was a patient with diarrhea combined with a positive stool for CD toxin or for a toxigenic CD identified by polymerase chain reaction (PCR).

Only unformed fecal samples following the Bristol stool scale were accepted for testing, after the patient’s medical history was reviewed. From 2009 to 2012, the diagnostic technique was a combined EIA, the C. DIFF QUIK CHEK COMPLETE^®^ test (Abbott Laboratories, Chicago, IL, USA), which detects CD-specific GDH and CD toxin A/B in the fecal specimen. Starting in 2013, a real-time PCR (GenomEra^®^ CD, Abacus Diagnostica, Turku, Finland) was added as the second step in a two-step screening protocol. Samples that were GDH-positive but toxin-negative were retested by PCR in order to detect the presence of the *tcdB* gene that encodes the tcdB protein in a toxigenic CD strain.

Treatment of CDI followed hospital guidelines based on the European Society of Clinical Microbiology and Infectious Diseases (ESCMID) consensus and remained unchanged during the study period. Oral metronidazole was given to non-severe cases and oral vancomycin to severe cases [8].

For each case, we completed a database that included the following variables: demographic characteristics, admission ward, place of acquisition of the infection, comorbidities, exposure to known factors associated with CDI in the last 30 days, symptoms, analytical parameters, diagnostic method, CDI treatment, outcomes and length of hospital stay (LOS). Data were obtained for all patients from their digital medical files.

Classification according to the site of acquisition of CDI: (1) Hospital-acquired CDI: infection identified >48 h after hospital admission and before discharge. (2) Non-nosocomial healthcare-related CDI: infection starting in the community or within 48 h of admission, in patients admitted to a health center (hospital, nursing home or community health center) in the 4 weeks prior to the onset of symptoms. (3) Community-acquired CDI: infection starting in the community or within 48 h of admission, with no admission to a health center in the last 4 weeks [12]. A case of severe CDI was defined when at least one of the following criteria was present: ≥15,000 white blood cells (WBC) per milliliter or >50% above the baseline increase in serum creatinine. As per the guidelines definition, a case of complicated CDI was defined when at least one of the following was present: circulatory shock, ileus, <2000 or ≥35,000 WBC per milliliter or >2.2 micromoles of serum lactate per milliliter [8]. A case was defined as recurrent CDI if a new episode of diarrhea with a positive stool test occurred within 8 weeks after the resolution of the previous episode with at least 10 days of treatment. Crude mortality was defined as death from any cause within 30 days of the diagnosis of CDI.

### 2.2. Patients Design Statement

The design of the work has been approved by the Clinical Ethics Committee of the hospital (PI-18-150); the need for informed consent was waived due to the retrospective nature of the study.

### 2.3. Statistical Analysis

Incidence density of CDI was calculated as the number of CDI diagnosed at the hospital per 10,000 hospital-days. The incidence density of tested stool samples for CD was calculated as the number of tested stool samples per 10,000 hospital-days. The rate of CD test positivity was calculated as the percentage of positive tests relative to the total number of tests performed. The incidence density of CDI and stool samples tested and positive samples for CD were calculated overall and per year to evaluate temporal trends. A Poisson regression model was used to assess longitudinal trends.

Quantitative variables were described in terms of mean and standard deviation. Categorical variables were expressed in absolute frequencies and percentages. A univariate analysis of the relationship between the variables and CD diagnostic test, mortality or recurrence was performed using the Student’s *t*-test for continuous data and the chi-square test for categorical variables. A logistic regression analysis adjusted for age and sex was performed, entering the variables that were significant on univariate analysis. SPSS Statistics 25 (IBM; Armonk, NY, USA) was used.

## 3. Results

A total of 443 patients with a first episode of CDI were included, of which 297 (67.1%) were toxin-positive and the remaining 146 (32.9%) were identified thanks to the two-step algorithm including the PCR test in GDH positive/toxin negative cases. Seventy-five cases were diagnosed in the first time period before the introduction of the algorithm (2009–2012) while 368 cases were detected after its introduction. (2013–2018).

### 3.1. CDI Incidence along the Time Periods

The overall mean incidence of CDI over the period 2009–2018 was 3.12 cases per 10,000 patient-days with a significant increase in incidence from 0.9 in 2009 to 5.5 in 2018 (*p* < 0.01) (Figure 1).

This increase was observed regardless of CDI origin: hospital-acquired infections rose from 0.66 to 2.11, community-acquired from 0.12 to 0.83 and non-hospital healthcare-acquired from 0.24 to 0.60 (*p* < 0.01 for all).

The overall testing rate for CDI was 46.1 tests per 10,000 patient-days, with a significant increase from 29.6 tests in 2009 to 65.7 per 10,000 patient-days in 2018 (*p* < 0.01). Moreover, the mean percentage of positive tests for CDI in the study period was 8.5%, with a significant increase from 3.9% in 2009 to 11.2% in 2018 (*p* < 0.01).

### 3.2. Characteristics of the Patients

The origin of CDI was hospital-acquired in 223 (50.3%) cases, healthcare-related in 117 (26.4%) and community-acquired in 103 (23.3%). Demographics, risk factors, clinical presentation, treatment and outcomes are shown in Table 1.

A total of 401 (90.5%) patients were hospitalized, of which 332 (82.7%) were admitted to the medical department, 41 (10.2%) to the surgical department and 36 (8.9%) to the intensive care unit (ICU). The mean length of hospital stay was 28.6 ± 27.4 days and 76 patients (18.9%) were discharged to a long-term care facility (LTCF). The rate of discharge to an LTCF was higher in patients with healthcare-associated CDI (non-hospital healthcare-related or hospital-acquired CDI) compared with those with community-acquired CDI (30.3 vs. 10.6%, *p* = 0.02).

Characteristics and outcomes regarding the CDI diagnostic method are shown in Table 2. Severe presentation (OR 2.1; 95% CI: 1.26–3.55) was significantly associated with CD toxin-positive while inflammatory bowel disease (IBD) (OR 7.2; CI 95% CI: 1.90–27.26) and immunosuppressive treatment (OR 2.5; 95% CI: 1.34–4.62) were significantly associated with toxin-negative/PCR-positive.

### 3.3. CDI Treatment

A total of 437 (98.6%) patients received a specific antibiotic for CDI: metronidazole in monotherapy in 312 (70.4%) and vancomycin in 122 (27.5%) cases, in monotherapy in 86 cases or associated with metronidazole in 36 cases. Fidaxomicin was only administered in three cases. The mean duration of treatment was 13.3 ± 5.6 days. The use of vancomycin increased from 20% of cases in 2009 to 43.2% in 2018 (*p* = 0.09). Patients with severe (56.6% vs. 33.9%, *p* < 0.001) or complicated CDI (27% vs. 10.4%, *p* < 0.01) more frequently received vancomycin than metronidazole, in accordance with guideline recommendations. Non-CDI dispensable antibiotics were discontinued in 154 (34.8%) cases and proton-pump inhibitors (PPI) in 26 (5.9%).

### 3.4. Clinical Outcomes

A total of 63 (14.2%) patients died within 30 days of the CDI diagnosis. On multivariate analysis, hematological neoplasm (OR 2.8; 95%CI: 1.19–6.59), intensive care unit (ICU) admission (OR 5.4; 95%CI: 2.35–12.61) and complicated CDI (OR 2.8; 95%CI: 1.42–5.71) were associated with increased mortality (Table 3).

The rate of CDI recurrence was 14.9%, and exposure to PPI (OR 2.29; 95%CI: 1.1–4.7) and severe previous presentation (OR 1.79; 95%CI: 1.03–3.12, *p* = 0.03) were significantly associated with increased risks on multivariate analysis (Table 4).

## 4. Discussion

The results of this study show a sixfold increase in the incidence of CDI at our center from 2009 to 2018, coinciding with a significant augment in the testing rate and in the percentage of positive stools tested after the introduction of an optimized two-step diagnostic algorithm.

The incidence of patients diagnosed with a first episode of CDI in our center increased significantly from 0.9 to 5.5 cases per 10,000 patient-days and it is consistent with previous studies [4,7,10]. Several factors may explain this increase in the incidence of CDI in addition to the introduction of the more sensitive two-step diagnostic protocol in 2013 [7,10,11,13]. These factors include an augment in the population at risk and a higher index of clinical suspicion. Hospitalization is another important risk factor for CDI, since highly susceptible patients are exposed to a spore-contaminated environment when more cases are present. The incidence of CDI achieved in 2018 (5.5 per 10,000 patient-days) is similar to the rates reported in other Mediterranean countries [7,8,10], but lower than those reported from northern Europe [4,6], the United States [1] or Canada [10,12] (7.9 to 16.2 cases per 10,000 patient-days). However, we observed a plateau in the incidence during the last two years of the study (2016–2018). The reasons for this stabilization in the more recent years fall out of the scope of the present analysis but are consistent with a recent study showing a decline in CDI in the United States from 2011 to 2018 due to a fall in health-care and hospital-acquired infections [4].

Importantly, one-third of patients with CDI diagnosed using the two-step algorithm would have been missed or diagnosed later if only EIA-toxin testing had been performed [9]. An earlier diagnosis allows for quick onset of targeted treatment, prompt correction of predisposing factors and could improve the prognosis of the disease. Moreover, considering that patients who are PCR-positive only present a risk of transmission similar to those who test toxin-positive, an earlier detection of CDI cases is important to prevent nosocomial transmission [14]. Concerns about the detection of colonized patients with a positive PCR were unlikely a problem in our study because only patients with diarrhea were tested by the microbiology laboratory after the patient’s medical history was reviewed. Therefore, all PCR-positive cases were symptomatic.

Regarding patient characteristics, there was a high prevalence of known risk factors for CDI, such as older age, health-care origin and exposure to antimicrobials and PPI [2,7]. It should be noted that nearly half of the patients had some kind of immunosuppression such as solid or hematological neoplasm, HIV infection, IBD, immunosuppressive therapy or chemotherapy [15,16]. In general, immunocompromised patients have a higher incidence of CDI and recurrence, probably due to their own immunosuppression, higher antimicrobial use, increased exposure to healthcare settings and higher prevalence of CD colonization [11,13].

The origin of the CDI in a quarter of our series was community acquired, a finding that is comparable to other studies [17,18]. Though this incidence was still lower than that of healthcare-associated CDI, it increased over the period of study. This has been attributed in other countries to the greater awareness among physicians of the possibility of CDI with no hospital exposure, and probably also by the increased use of antibiotics and proton-pump inhibitors in the community [5].

We identified that CD toxin-positive patients have a more severe clinical presentation. This finding is in agreement with previous reports [19,20,21,22]. The most likely explanation is an increased CD bacterial load and higher toxin concentration than those toxin-negative diagnosed only by more sensitive methods such as a PCR [14].

The attending physicians should therefore be particularly aware when the patient is diagnosed with CDI by means of a positive toxin EIA in order to optimize treatment [21].

An unexpected finding was that PCR-diagnosed patients more frequently were receiving immunosuppressive drugs or had an IBD, perhaps due to the higher suspicion among gastroenterologists and the common practice of proactively testing for CDI in these fields [15,16,20]. Contrary to other studies, there were no differences in mortality and recurrences between the two groups [15,16,17,18,19].

Regarding antibiotic treatment of the first episode of CDI, metronidazole was the treatment most frequently prescribed; although, vancomycin was preferred for severe and complicated cases in accordance with guideline recommendations [12]. Coinciding with previous studies, there was a progressive increasing use of vancomycin along the study time period, probably due to the increase seen in the severity of cases [20]. Fidaxomicin was only used in a few cases due to its restricted use due to its cost. Metronidazole could still be an alternative for non-severe CDI with low risk of recurrence when the above agents are unavailable [23].

Crude 30-day all-cause mortality (14.2%) was similar to that reported in other studies [19,21,22]. As previously described, mortality was significantly higher in patients admitted to ICU and in those with complicated diseases [4,24,25,26]. Patients with hematological neoplasms also presented a higher mortality, making them a particular target for improved treatment and prevention of CDI [23]. Other factors previously reported such as older age, non-CDI antibiotic use and the presence of underlying comorbid conditions were not associated with increased mortality in multivariate analysis in our study [8,27].

In addition, as previously described, patients hospitalized with CDI had a higher probability of being discharged to an LTCF, mainly those with nosocomial and healthcare-associated infections [27]. Several risk factors for a non-home discharge include comorbidity and aging.

The rate of recurrence (14.9%) was comparable to other studies, with rates ranging from 12% to 30% [18,27,28]. Consistent with other investigations, recurrence was more common in patients with previous exposure to PPI and severe presentation [1]. However, other known risk factors associated with recurrence in other studies such as advanced age, chronic renal insufficiency, elevated white blood cell counts, low serum albumin levels and continued use of systemic antimicrobials during the initial CDI episode were not significantly associated with a higher recurrence rate in our multivariate analysis [28].

Our study has several limitations. A weakness is the retrospective design, which may have introduced selection bias and could imply information loss relative to the variables studied. The increase in CDI diagnosis could be associated with other uncontrolled factors in addition to the two-step lab diagnostic algorithm, such as increased physician awareness or increased rates of immunosuppressed individuals or aged patients.

## 5. Conclusions

The incidence of CDI at our center increased significantly over the decade comprising this study period after the introduction of a two-step laboratory diagnosis algorithm including a PCR. This occurred simultaneously with increased rates of testing and positivity in the samples. A third of the cases diagnosed with the new algorithm would have been missed using only a toxin-based screening strategy. This earlier diagnosis can have an impact on patient outcomes and hospital transmission. CD toxin-positive patients had a more severe clinical presentation, probably due to an increased CD bacterial load with higher toxin concentration. Therefore, a positive CD toxin must alert clinicians about the worse prognosis of the case.

## Figures and Tables

**Figure 1 microorganisms-10-01075-f001:**
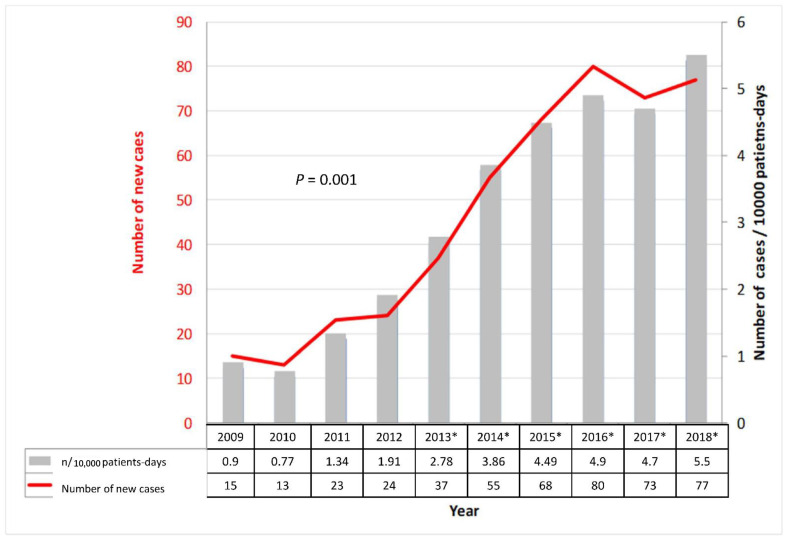
Trends in the number of new cases and incidence density of CDI (2009–2018). * PCR was added as the second step in the two-step screening protocol.

**Table 1 microorganisms-10-01075-t001:** Demographics, risk factors and clinical characteristics of patients included in the study (N = 443 cases).

Characteristic	N (%)
Age, mean (year, SD)	68.2 ± 16.3
Male gender	241 (54.4)
Risk factors	
Chronic pulmonary disease	95 (21.4)
Chronic renal disease	119 (26.9)
Diabetes mellitus	127 (28.7)
Heart failure	99 (22.3)
Solid organ cancer	136 (30.7)
Hematologic neoplasm	47 (10.6)
Liver disease	53 (12)
Inflammatory bowel disease	21 (4.7)
HIV infection	14 (3.2)
Charlson index ≥ 3	164 (37.1)
Abdominal surgery	19 (4.3)
Chemotherapy	50 (11.3)
Enteral nutrition	53 (12)
Immunosuppressive drugs	82 (18.5)
Non-CDI antibiotic use	348 (78.6)
PPI use	304 (68.6)
Clinical and analytical presentation	
Abdominal pain	206 (46.5)
Fever	108 (24.4)
Shock	33 (7.4)
Severe case	174 (39.7)
Complicated case	66 (15)
Leukocytes > 15,000 cells/mm^3^	110 (25.2)
Albumin (g/L, SD)	27.8 ± 6.5
Creatinine (mg/dL, SD)	1.7 ± 1.7
Hemoglobin (g/dL, SD)	10.7 ± 2.1

SD: standard deviation; HIV: human immunodeficiency infection; CDI: *Clostridioides difficile* infection; PPI: proton pump inhibitor.

**Table 2 microorganisms-10-01075-t002:** Univariate and multivariate predictors of toxin-positive CDI.

Characteristic	Toxin + N = 297 (67%) N (%)	Toxin −/PCR + N = 146 (33%) N (%)	*p* Value (Unadjusted Analysis)	*p* Value (Adjusted Analysis)
Age, mean (SD)	69.8 ± 16.1	64.9 ± 16.3	0.003	
Male gender	155 (52.2)	86 (58.9)	0.21	
Chronic pulmonary disease	69 (23.2)	26 (17.8)	0.23	
Chronic renal disease	83 (27.9)	36 (24.7)	0.53	
Diabetes mellitus	85 (28.6)	42 (28.8)	1	
Heart failure	75 (25.3)	24 (16.4)	0.05	
Solid organ cancer	83 (27.9)	53 (36.3)	0.09	
Hematologic neoplasm	27 (9.1)	20 13.7)	0.18	
Liver disease	28 (9.4)	25 (17.1)	0.03	
Inflammatory bowel disease	3 (1)	18 (12.3)	<0.001	0.004
HIV infection	7 (2.4)	7 (4.8)	0.27	
Charlson index ≥ 3	107 (36.1)	57(39)	0.21	
Abdominal surgery	11 (3.7)	8 (5.5)	0.53	
Chemotherapy	28 (9.4)	22 (15.1)	0.07	
Enteral nutrition	30 (10.1)	23 (15.8)	0.11	
Immunosuppressive drugs	43 (14.5)	39 (26.7)	0.002	0.004
Non-CDI antibiotic use	244 (82.2)	104 (71.2)	0.01	
Proton pump inhibitor use	217 (73.1)	87 (59.6)	0.006	
Abdominal pain	150 (50.5)	56 (38.4)	0.01	
Fever	73 (24.6)	35 (24)	0.98	
Shock	28 (9.4)	5 (3.4)	0.03	
Leukocytes > 15,000 cells/mm^3^	91 (31.2)	19 (13.2)	<0.001	
Albumin (g/L, SD)	27.6 ± 6.6	28.1 ± 6.2	0.48	
Creatinine (mg/dL, SD)	1.7 ± 1.7	1.4 ± 1.4	0.07	
Hemoglobin (g/dL, SD)	10.7 ± 1.9	10.5 ± 2.3	0.37	
Severe presentation	133 (45.2)	41 (28.5)	0.001	0.004
Complicated case	48 (16.3)	18 (12.4)	0.35	
Non-CDI antibiotic suppression	118 (40)	36 (25.4)	0.004	
PPI suppression	19 (6.4)	7 (4.9)	0.66	
Recurrence	52 (17.6)	14 (9.6)	0.04	
Death (30d)	42 (14.2)	21 (14.4)	0.95	

SD: standard deviation; HIV: human immunodeficiency infection; CDI: *Clostridioides difficile* infection; PCR: polymerase chain reaction; PPI: proton pump inhibitor.

**Table 3 microorganisms-10-01075-t003:** Univariate and multivariate predictors of death (30 d).

Characteristics	Deaths (30 d) N = 63 N (%)	Cured N = 380 N (%)	*p* Value (Unadjusted Analysis)	*p* Value (Adjusted Analysis)
Age, mean (SD)	69.8 ± 17.7	67.9 ± 16.1	0.39	
Male gender	32 (50.8)	209 (55)	0.60	
ICU admission	14 (22.6)	22(6.5)	<0.001	<0.001
Chronic pulmonary disease	16 (25.4)	79 (20.8)	0.51	
Chronic renal disease	20 (31.7)	99 (26.1)	0.42	
Diabetes mellitus	23 (36.5)	104 (27.4)	0.18	
Heart failure	16 (25.4)	83 (21.8)	0.64	
Solid organ cancer	15 (23.8)	121 (31.8)	0.25	
Hematologic neoplasm	12 (19)	35 (9.2)	0.03	0.01
Liver disease	10 (15.9)	43 (11.3)	0.41	
Inflammatory bowel disease	3 (4.8)	18 (4.7)	1	
HIV infection	4 (6.3)	10 (2.6)	0.12	
Charlson index ≥ 3	24 (38.1)	140 (36.8)	0.96	
Abdominal surgery	1 (1.6)	18 (4.7)	0.49	
Chemotherapy	6 (9.5)	43 (11.3)	1	
Enteral nutrition	9 (14.3)	44 (11.6)	0.68	
Immunosuppressive drugs	17 (27)	65 (17.1)	0.09	
Non-CDI antibiotic use	55 (87.3)	293 (77.1)	0.09	
Proton pump inhibitor use	39 (61.9)	265 (69.7)	0.27	
Abdominal pain	26 (41.3)	180 (47.4)	0.44	
Fever	15 (23.8)	93 (24.5)	1	
Shock	12 (19)	21 (5.5)	<0.001	
Ileus	2 (3.1)	2 (0.5)	0.09	
Toxic megacolon	2 (3.2)	0	0.02	
Leukocytes > 15,000 cells/mm^3^	26 (42.6)	64 (22.5)	0.001	
Albumin (g/L, SD)	23.6 ± 6.5	28.5 ± 6.1	<0.001	
Creatinine (mg/dL, SD)	1.7 ± 1.5	1.6 ± 1.6	0.53	
Hemoglobin (g/dL, SD)	10.2 ± 2.1	10.7 ± 2.1	0.08	
Severe presentation	33 (53.2)	146 (38.4)	0.05	
Complicated case	20 (32.3)	46 (12.2)	<0.001	0.003
CD Toxin-positive	42 (66.7)	255 (67.1)	1	
Vancomycin ± metronidazole	24 (38.1)	98 (26.4)	0.79	
Non-CDI antibiotic suppression	15 (24.6)	139 (37)	0.08	
PPI suppression	2 (3.2)	24 (6.3)	0.55	

SD: standard deviation; ICU: intensive care unit; HIV: human immunodeficiency infection; CDI: *Clostridioides difficile* infection; PCR: polymerase chain reaction; PPI: proton pump inhibitor; *Clostridioides difficile.*

**Table 4 microorganisms-10-01075-t004:** Univariate and multivariate predictors of recurrences.

Characteristic	Recurrence N = 66 N (%)	Non-Recurrence N = 343 N (%)	*p* Value (Unadjusted Analysis)	*p* Value (Adjusted Analysis)
Age, mean (SD)	71.9 ± 12.8	67.1 ± 16.6	0.05	
Male gender	36 (54.5)	173 (55.1)	1	
UCI admission	4 (6.7)	18 (6.4)	1	
Chronic pulmonary disease	17 (25.8)	62 (19.7)	0.27	
Chronic renal disease	23 (34.8)	76 (24.2)	0.07	
Diabetes mellitus	22 (33.3)	82 (26.1)	0.23	
Heart failure	20 (30.3)	63 (20.1)	0.06	
Solid organ cancer	18 (27.3)	103 (32.8)	0.38	
Hematologic neoplasm	6 (9.1)	29 (9.2)	0.97	
Liver disease	9 (13.6)	34 (10.8)	0.51	
Inflammatory bowel disease	2 (3)	16 (5.1)	0.75	
VIH infection	3 (4.5)	7 (2.2)	0.38	
Charlson index ≥ 3	28 (42.4)	112 (35.7)	0.30	
Abdominal surgery	5 (7.6)	13 (4.1)	0.23	
Chemotherapy	1 (1.5)	42 (13.1)	0.003	
Enteral nutrition	8 (12.1)	36 (11.5)	0.88	
Immunosuppressive drugs	10 (15.2)	55 (17.5)	0.64	
Non-CDI antibiotic use	54 (81.8)	239 (76.1)	0.31	
Proton pump inhibitor use	56 (84.8)	209 (66.6)	0.003	0.02
Abdominal pain	36 (54.5)	144 (45.9)	0.19	
Fever	16 (24.2)	77 (24.5)	0.96	
Leukocytes > 15,000 cells/mm^3^	24 (36.9)	60 (19.4)	0.002	
Albumin (g/L, SD)	28.1 ± 4.8	28.6 ± 6.4	0.59	
Creatinine (mg/dL, SD)	2.2 ± 2.3	1.4 ± 1.4	0.001	
Hemoglobin (g/dL, SD)	10.5 ± 1.6	10.7 ± 2.1	0.52	
Severe presentation	35 (53)	111 (35.4)	0.007	0.03
Complicated case	6 (9.1)	40 (12.7)	0.40	
CD toxin-positive	52 (78.8)	203 (64.6)	0.02	
Vancomycin ± metronidazole	19 (29.7)	79 (25.7)	0.51	
Days of treatment (mean, SD)	14.8 ± 7.4	13.7 ± 4.6	0.27	
Non-CDI antibiotic suppression	27 (42.2)	112 (35.9)	0.34	
PPI suppression	6 (9.1)	18 (35.9)	0.34	

SD: standard deviation; ICU: intensive care unit; HIV: human immunodeficiency infection; CDI: *Clostridioides difficile* infection; PCR: polymerase chain reaction; PPI: proton pump inhibitor; CD: *Clostridioides difficile.*

## Data Availability

The data collected for this study can be shared upon a justified request and following the approval of the Ethics Committee of the University Hospital Germans Trias i Pujol, Badalona, Spain. Those interested should address the request to Nieves Sopena: nsopena.germanstrias@gencat.cat.

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
