# Peer review of "Impact of the Introduction of a Two-Step Laboratory Diagnostic Algorithm in the Incidence and Earlier Diagnosis of Clostridioides difficile Infection"

_microorganisms, 2022, doi:10.3390/microorganisms10051075_

Round 1

Reviewer 1 Report

The following comments may help the authors to improve the manuscript.

  1. Figure 1, the authors should edit this figure to show the labels at the x and y axes.
  2. Would there be a method to reduce the number of tests at the central laboratories, e.g. testing at home (point of care tests)?

Author Response

 1. Figure 1, the authors should edit this figure to show the labels at the x and y axes.

Thanks for noting this. We have edited Figure 1 to show the labels at the x and y axes.

 2. Would there be a method to reduce the number of tests at the central laboratories, e.g. testing at home (point of care tests)?

There are several studies using diagnostic test for CDI as POCT in settings with slow response of the central laboratory.  However, to our knowledge there have not been used at home. In addition, the response of our Lab is ususally quite fast, so we profer to stick on our validated lab.

Reviewer 2 Report

Clostridioides difficile becomes an agent more and more encountered in the hospital today. Careful diagnosis is an important step for defining the treatment. The presented papers presents in the title the consequences of the introduction of a second laboratory test on the incidence and clinical outcome of this infection. But the following presentation of the results does not really concern what is mentionned in the title. The results present the clinical data of all patients encountered during ten years both before and after the introduction of the second test. The paper needs to be completely revised to meet the title.

Minor remarks:

  • C. difficile survives in the environment. The increase in incidence (as mentionned in the introduction) can also be due to a heavier contamination of the environment when more cases are present.
  • The incidence of these infection is increasing all over the world. So the increase stated in this hospital seems not exclusively link to a better diagnosis (page 2).
  • In figure 1 it is not indicated what the asterisk indicates (even if we understand that it concerns the introduction of the real-time PCR test).

Author Response

Clostridioides difficile becomes an agent more and more encountered in the hospital today. Careful diagnosis is an important step for defining the treatment. The presented papers presents in the title the consequences of the introduction of a second laboratory test on the incidence and clinical outcome of this infection. But the following presentation of the results does not really concern what is mentionned in the title. The results present the clinical data of all patients encountered during ten years both before and after the introduction of the second test. The paper needs to be completely revised to meet the title.

We agree with the reviewer that the title does not accurately reflect the results and conclusions of the study. We have modified the title accordingly, that now reads “Impact of the introduction of a two-step laboratory diagnostic algorithm in the incidence and earlier diagnosis of Clostridioides difficile infection.

In addition, we have modified a sentence in the implications of the study to a conditional verb in the abstract.

Minor remarks:

  • C. difficile survives in the environment. The increase in incidence (as mentioned in the introduction) can also be due to a heavier contamination of the environment when more cases are present.

We agree with this comment. We have added the following sentence to clarify this issue: “Hospitalization is another important risk factor for CDI, since highly susceptible patients are exposed to spore-contaminated environment when more cases are present.”

  • The incidence of this infection is increasing all over the world. So the increase stated in this hospital seems not exclusively link to a better diagnosis (page 2).

We agree with this comment. In the introduction it states “The incidence of CDI has increased in the last decade, probably due to the increasing numbers of patients at risk and improvements in diagnostic methods”.

In the discussion, we mention “Several factors may explain this increase in the incidence of CDI in addition to the introduction of the more sensitive two-step diagnostic protocol, including an augment in the population at risk and a higher index of clinical suspicion.”

No further changes have been implemented regarding this issue. 

  • In figure 1 it is not indicated what the asterisk indicates (even if we understand that it concerns the introduction of the real-time PCR test).

The reviewer is right. We have included as a footnote “*PCR was added as the second step in the two-step screening protocol” in Figure 1.

Round 2

Reviewer 1 Report

I do not have further comments

Author Response

Reviewer 1

I do not have further comments.

We are happy that the changes implemented satisfy the reviewer’s suggestions.

Reviewer 2 Report

Thank you for submitting a new version of your paper. But the high discrepancies between the aim mentionned in the title and the results presented are still existing. The aim of your paper is not to characterize the patients suffering from CD infection but to define the differences before and after the introduction of the second step.

Author Response

Thank you for submitting a new version of your paper. But the high discrepancies between the aim mentioned in the title and the results presented are still existing. The aim of your paper is not to characterize the patients suffering from CD infection but to define the differences before and after the introduction of the second step.

We agree that the study was not designed to analyze the clinical changes in CDI cases before and after the introduction of the two-step diagnosis algorithm.

The main aim of our study has been to analyze the changes in the incidence of CDI over a decade with the introduction of a two-step laboratory diagnosis algorithm. We have also analyzed the epidemiological characteristics, risk factors, clinical presentation, diagnostic test used, treatment received and outcomes (recurrences and mortality) of all CDI patients diagnosed along this period.

The study results confirm that 146 (32.9%) cases were only diagnosed through the two-step algorithm, as they were EIA negative and GDH positive. As in the previous period all EIA negative cases were not further studied and were not considered as CDI, the two-step algorithm captures up to one third CDI cases that would have been missed in the prior period. We speculate that this earlier diagnosis could potentially (and we emphasize potentially because this has not been proven) reduce nosocomial transmission and could positively impact patient outcomes.

We have modified accordingly the title, the abstract, the main study aim in the introduction, and clarified the incidence density in methods.

We really hope that all these changes make the study clearer for the journal readers.